# An Adaptive Median Filter Based on Sampling Rate for R-Peak Detection and Major-Arrhythmia Analysis

**DOI:** 10.3390/s20216144

**Published:** 2020-10-29

**Authors:** Tae Wuk Bae, Sang Hag Lee, Kee Koo Kwon

**Affiliations:** 1Daegu-Gyeongbuk Research Center, Electronics and Telecommunications Research Institute, Daegu 42994, Korea; kwonkk@etri.re.kr; 2TriBell Labs, Kyungpuk 38541, Korea; lsh6465@tribell-lab.com

**Keywords:** adaptive median, sampling rate, R peak, arrhythmia, heartrate variability

## Abstract

With the advancement of the Internet of Medical Things technology, many vital sign-sensing devices are being developed. Among the diverse healthcare devices, portable electrocardiogram (ECG) measuring devices are being developed most actively with the recent development of sensor technology. These ECG measuring devices use different sampling rates according to the hardware conditions, which is the first variable to consider in the development of ECG analysis technology. Herein, we propose an R-point detection method using an adaptive median filter based on the sampling rate and analyze major arrhythmias using the signal characteristics. First, the sliding window and median filter size are determined according to the set sampling rate, and a wider median filter is applied to the QRS section with high variance within the sliding window. Then, the R point is detected by subtracting the filtered signal from the original signal. Methods for detecting major arrhythmias using the detected R point are proposed. Different types of ECG signals were used for a simulation, including ECG signals from the MIT-BIH arrhythmia database and MIT-BIH atrial fibrillation database, signals generated by a simulator, and actual measured signals with different sampling rates. The experimental results indicated the effectiveness of the proposed R-point detection method and arrhythmia analysis technique.

## 1. Introduction

The development of wearable medical devices has accelerated with the advancement of sensor technology. Among the various biometric parameters, the electrocardiogram (ECG) is the most important biosignal. To date, diverse wearable and portable ECG devices have been commercialized. The Seers Tech. Biopatch is a wearable ambulatory cardiac monitoring device with a single lead and a sampling rate of 256 samples/s [1]. KardiaMobile from AliveCor is a finger-contact mini ECG measurement tool with a sampling rate of 300 samples/s [2]. The recently developed TLC5000 made by Contect is a 12-lead ECG system with a sampling rate of 10,000 samples/s [3]. The currently available ECG devices have different sampling rates, which is an important variable to be considered when developing an ECG analysis algorithm.

Likewise, public databases were created at various sampling rates, e.g., the MIT-BIH arrhythmia database (MITDB) at 360 Hz [4], MIT-BIH atrial fibrillation database (AFDB) at 250 Hz [5], QT database [6], and Long Term AF database at 128 Hz [7]. The Contec MS400—an ECG multiparameter simulator—was used to obtain reference arrhythmia signals at 250 Hz [8].

In previous works, R-point detection methods using the sampling frequency-based Hilbert transform [9] and the sampling period-based difference equation [10,11] were proposed. However, several R-point detection studies have been performed at specific sampling rates. The linear adaptive filter [12], three-point center derivative [13], morphology and differential filter [14], max–median–min filter [15], generic algorithm [16], wavelet transform [17], wavelet denoising [18], multiscale mathematical morphology (3M) [19], and derivative-max filter [20] use predefined sizes for filter or window selection without considering the sampling rate. And recently, QRS detection [21] and T-wave detection [22] using adaptive filters have been proposed.

Typically, ECGs are used for arrhythmia detection and prediction or in emergency care situations. Recently, ECG measuring devices and telemedicine systems have been actively studied. Using Bluetooth, communication mobile telephony network, and wireless local area network technologies, recent telemedicine systems measure and transmit ECG data in real-time to protect heart-disease patients [23,24,25]. Occlusion in one of the coronary arteries of the heart leads to cardiac ailment and myocardial infarction (MI). The localization of MI based on the investigation of the morphology of the multi-lead ECG is the initial task for the diagnosis of this ailment [26]. These telemedicine systems for constant observation of coronary heart disease patients provide the possibility for specialists to interpret ECGs on mobile devices, bridging the gap between patients and specialists [27]. The automated external defibrillator, which is used urgently in the event of a heart attack, analyzes the patient’s heart rhythm by measuring ECG signals to perform accurate defibrillation before the defibrillation. Additionally, after cardiac resynchronization therapy, ECG signals are measured to analyze the patient’s ECG pattern [28,29]. The sampling rate has a significant influence on the filter design or preprocessing for R-point detection. If the sampling rate is low, the calculation speed is high, but it is difficult to obtain detailed information, such as the fragmented R peak and the degree of noise. Conversely, when the sampling rate is high, the processing speed is low because of the large filter size, but it is easy to extract accurate information for each ECG wave. Thus, different ECG devices have different sampling rates, and the R-point detection algorithm must cope with various sampling rates.

The detected R points are important for detecting cardiac arrhythmia. To date, arrhythmia studies have been conducted with a bias toward specific arrhythmias, e.g., atrial fibrillation (AF) or ischemia. Although bigeminy and trigeminy with different R periods due to ectopic beats or premature ventricular contraction (PVC) may progress to dangerous arrhythmia, few related studies have been performed [30,31]. Although PVCs caused by ventricular extra heartbeats are dangerous when they exceed 20% of the total heartbeat, only detection studies based on signal processing have been performed; no studies have involved heartrate variability (HRV) analysis of ventricular tachycardia (VT) with PVCs [32,33]. Supraventricular tachycardia (SVT)—an abnormally fast heartbeat—may be a precursor to ventricular fibrillation, but studies on this topic are not actively performed [34,35]. Therefore, there is a need for continued research on the aforementioned cardiac arrhythmias as well as the well-known AF.

For the foregoing reasons, it is necessary to develop a universal ECG analysis algorithm to analyze ECG signals having different sampling rates obtained from various ECG measuring devices. This paper proposes an R-point detection method using an adaptive median filter in which the sliding window and filter size are automatically adjusted according to the sampling rate of the measuring device. Additionally, this paper introduces HRV analysis of not only AF, which is widely studied, but also bigeminy, trigeminy, PVC, VT, and SVT, which are not actively studied, based on the signal characteristics.

## 2. Materials and Methods

### 2.1. ECG Data Used

Four types of ECG data (MITDB [4], AFDB [5], actually measured ECG signals, and signals generated by an MS400 multiparameter simulator [8]) were used, as shown in Figure 1. The MITDB contained 48 half-hour excerpts of two-channel ambulatory ECG recordings obtained from 47 subjects studied by the BIH Arrhythmia Laboratory between 1975 and 1979. The recordings were digitized at 360 samples per second per channel with an 11-bit resolution over a 10-mV range. MITDB is used to evaluate the R point detection method proposed in this paper. The AFDB included 25 long-term ECG recordings of human subjects with AF (mostly paroxysmal). The individual recordings each had a duration of 10 h and contained two ECG signals (sampled at 250 samples/s) with 12-bit resolution over a range of ±10 mV. AFDB is used to test the AF detection technique proposed in this paper and visualize the severity of the test AF signals. Actual ECG data were measured using a VP-100 (ECG measurement patch), which was developed by TriBell-Lab [36] and ETRI (Electronics and Telecommunications Research Institute), at 250 samples/s. The bigeminy signal used in this paper was measured by the developed ECG patch. Arrhythmias that were difficult to measure were generated by the MS400 simulator.

### 2.2. R-Peak Detection

The filtered signal preserving P or T waves while restraining the R wave for an original signal *S* is obtained by applying the following median filter [37]:(1)$$M_{i,n}={\mathrm{median}}_{r\in W}\left(S_{i+r,n}\right)$$
where the subscript (*i*,*n*) denotes the *i*th sample point location of the *n*th sliding window, and *W* represents the filter window. In the proposed method, a median filter with a variable filter size is used to improve the performance of a general median filter [38,39]. Figure 2 shows the relationship between the variance at the sample point and the median-filter size for AFDB of 250 samples/s. A median filter with a large filter size is applied to QRS interval having a high variance, while a median filter with a small filter size is applied to flat region (or weak wave region) among non-QRS interval having a low variance. Also, a median filter with a basic size is applied to the P or T wave region among the non-qrs region corresponding to the moderate variance.

The adaptive filter size *r* is determined by the variance of the current sample point, as follows:(2)$$\begin{array}{l} \left\{ \begin{array}{cc} r=L-C, & v_{min}\left(S_{n}\right)\leq v\left(S_{i+r,n}\right)<D_{v}\left(S_{n}\right)/3\\ r=L, & D_{v}\left(S_{n}\right)/3\leq v\left(S_{i+r,n}\right)<D_{v}\left(S_{n}\right)\times 2/3\\ r=L+C, & D_{v}\left(S_{n}\right)\times 2/3\leq v\left(S_{i+r,n}\right)\leq v_{max}\left(S_{n}\right) \end{array}\right.\\ \mathrm{where}\;L=F/60,\;C=F/(60\times 2) \end{array}$$
where *L* represents the basic size of the filter window, which is determined by the sampling rate *F*. $D_{v}\left(S_{n}\right)$ represents the difference of $v_{max}\left(S_{n}\right)$ and $v_{min}\left(S_{n}\right)$, max, and min variance of the *n*th sliding window. The adaptive size of the median filter for a sampling rate of 250 Hz is shown in Figure 2. Additionally, the variance is obtained for the basic filter size. The variance section that determines the variable filter size is empirically divided into three uniform sections. The difference signal for the *n*th sliding window is given as follows:(3)$$D_{n}=\left|S_{n}-M_{n}\right|$$

The effectiveness of the adaptive median filter is shown in Figure 3, and the difference between the original and filtered signals was normalized. The adaptive median is compared with the normal median and exhibits a higher difference result, helping to accurately detect the R peak. A max filter with a filter size of *F*/8 is applied to the difference to obtain candidate intervals, including the R peak. The adaptive threshold for detecting the R peak in the candidate intervals is applied to the difference signal and is defined as the average of the top 50% of the maximum filter result.

Due to the absolute value characteristic of the difference signal in Equation (3), the R peak detected from the difference signal should be distinguished into a normal and abnormal R peak as follows:(4)$$\{\begin{array}{cc} \mathrm{Normal}\;\left(\mathrm{Positive}\right)\;R & \mathrm{if}\;\mathrm{detected}\;R>\mathrm{mean}\left(S_{n}\right)\\ \mathrm{Abnormal}\;\left(\mathrm{Negative}\;R\;\mathrm{or}\;\mathrm{PVC}\right) & \mathrm{else} \end{array}$$

If the detected R peak is greater than the average of the ECG signal of the current *n*th sliding window, it means a normal (positive) R, otherwise it is recognized as an abnormal R (negative R or PVC).

### 2.3. Detection of Important Arrhythmias

The processing units for various arrhythmias are presented in Table 1. It is possible to determine which beat is normal R or PVC with only one beat. However, bigeminy and trigeminy can be detected by multiple beats or 1 min long. Additionally, the identification of bradycardia, tachycardia, VT, SVT, and AF requires at least 1 min.

In the proposed method, the size of the sliding window is set according to the sampling rate at which the ECG signal is measured. By setting the sampling rate, the number of sliding windows per minute, the number of particle times constituting the sliding window, the time length of the particle time, the step size (movement length of sliding window), and the time length of the sliding window are automatically determined, as shown in Figure 4. The real-time processing of the proposed method involves a sliding window of 12 s, a step size of 8 s, an overlapped interval of 4 s, and seven sliding windows of 1 min:

The above sliding-window structure was derived manually to detect the arrhythmia presented in Table 1 in 1-min increments (every minute). Another sliding-window structure for detecting arrhythmia in units of 1 min may consist of Tp = 6 s, N_Tp = 4, and NSB = 3. Naturally, if the detection of arrhythmia in 1-min increments is ignored, various sliding-widget structures can be generated.

For a continuous ECG stream, the pseudocode for the detection of various arrhythmias is shown in Figure 5. According to the R peaks detected in a sliding window, Q onset, S offset, P peak, and T peak are searched for PVC, AF, VT, and SVT. The detection methods for various arrhythmias are introduced in the following sections.

#### 2.3.1. Bradycardia and Tachycardia 

Bradycardia is a very low heartrate (HR) of <60 beats per minute (BPM), and tachycardia is a very high HR of >100 BPM [40]. If tachycardia begins in the ventricles, it is called VT. Bradycardia and tachycardia can be easily detected by measuring the BPM.

#### 2.3.2. Bigeminy and Trigeminy

Bigeminy is a heart-rhythm condition involving repeated heartbeats of short and long cycles [41]. Trigeminy refers to a three-heartbeat pattern in which one or two beats are irregular [42]. In bigeminy, every other beat is a PVC. In trigeminy, every third beat is a PVC. Two consecutive PVCs are called couplets [43].

Figure 6 shows the beat-to-beat (RR) interval features of bigeminy and trigeminy. $\bar{{\mathrm{RR}}_{S}}$, $\bar{{\mathrm{RR}}_{M}}$, and $\bar{{\mathrm{RR}}_{L}}$ represent the average length of the short, middle, and long periods in observatin time, 1 min in this paper. While bigeminy has the feature that $\left(N\left({\mathrm{RR}}_{L}\right)≅N\left({\mathrm{RR}}_{S}\right)\right)\neq N\left({\mathrm{RR}}_{M}\right)$, trigeminy exhibits a characteristic that $N\left({\mathrm{RR}}_{L}\right)≅N\left({\mathrm{RR}}_{S}\right)≅N\left({\mathrm{RR}}_{M}\right)$. The average of the difference between the sums of the long and short periods $\bar{{\mathrm{RR}}_{\mathrm{LS}}}$ is given as follows:(5)$$\bar{{\mathrm{RR}}_{\mathrm{LS}}}=\left(\overset{t+T}{\underset{t}{\sum }}{\mathrm{RR}}_{L}-\overset{t+T}{\underset{t}{\sum }}{\mathrm{RR}}_{S}\right)/\left(\frac{N\left({\mathrm{RR}}_{L}\right)+N\left({\mathrm{RR}}_{S}\right)}{2}\right),$$
where *T* represents the observation time (in seconds). The formula for determining the bigeminy and trigeminy using Equation (4) is as follows: (6)$$\begin{array}{l} \mathrm{IF}\;{\mathrm{Th}}_{\mathrm{BG}\_\min}<\bar{{\mathrm{RR}}_{\mathrm{LS}}}<{\mathrm{Th}}_{\mathrm{BG}\_\max}\;\mathrm{and}\;\left|N\left({\mathrm{RR}}_{L}\right)-N\left({\mathrm{RR}}_{S}\right)\right|<{\mathrm{Diff}}_{\mathrm{LS}},\\ \;\mathrm{ECG}=\mathrm{Bigeminy}\\ \;\;\mathrm{IF}\;\bar{{\mathrm{RR}}_{S}}<\bar{{\mathrm{RR}}_{M}}\;\mathrm{and}\;\bar{{\mathrm{RR}}_{M}}<\bar{{\mathrm{RR}}_{L}}\;\mathrm{and}\;\left|N\left({\mathrm{RR}}_{M}\right)-N\left({\mathrm{RR}}_{S}\right)\right|<{\mathrm{Diff}}_{\mathrm{MS}},\\ \;\;\;\mathrm{ECG}=\mathrm{Trigeminy}\\ \;\;\mathrm{END}\\ \mathrm{END} \end{array}$$ where ${\mathrm{Th}}_{\mathrm{BG}\_\min}$ and ${\mathrm{Th}}_{\mathrm{BG}\_\max}$ represent the minimum and maximum values among the differences between the long and short periods for determining bigeminy.

#### 2.3.3. PVC, VT, and SVT

Figure 7 shows the features of PVCs, VT, and SVT. PVCs represent extra heartbeats that begin in one of two ventricles [44]. A PVC typically has a wide QRS interval (QRS width > 120 ms). In VT, the number of continuous PVCs is 3, 9, or greater, and typically the HR is ≥100 bpm. SVT includes many forms of heart arrhythmias that originate above the ventricles (or supraventricular) in the atria or AV node [45]. SVT is characterized by a very high HR between 150 and 250 bpm, as well as merged P and T waves, as shown in Figure 7. A PVC can be determined with just one beat, but the identification of VT and SVT requires at least 1 min of the signal. PVC, VT, and SVT can be detected by applying the aforementioned arrhythmia characteristics.

#### 2.3.4. AF

AF represents an arrhythmia characterized by fast and irregular beating of atrial chambers [46]. It often starts as short periods of abnormal beating, which increases over time. It begins as an atrial flutter, which then transforms into AF. AF is associated with an increased risk of heart failure, dementia, and stroke and is a type of SVT. It can be characterized by an irregular rhythm, the absence of a P wave and isoelectric baseline, and a variable ventricular rate. Its QRS complex is usually <120 ms, except in cases of a preexisting bundle branch block, an accessory pathway, or rate-related aberrant conduction [47]. Fibrillatory waves may be present and can be either fine (amplitude < 0.5 mm) or coarse (amplitude > 0.5 mm). Additionally, fibrillatory waves may mimic P waves, leading to misdiagnosis.

Figure 8 shows the five features used for AF detection. The proposed AF detection method is applied to an ECG signal length of ≥1 min to avoid the degradation of the feature detection accuracy due to noise and signal distortion. Because AF is characterized by no P wave or a very small P wave, it can be used as a feature for AF detection. The first feature for AF detection is the atrial activity (AA), which is measured as the P-wave area (PA). The P wave is defined by three points, i.e., the P-onset, P-peak, and P-offset, and the AA is measured according to the PA using the three P-wave related points. The second feature is the BPM, as AF has a high HR or BPM. Because the RR intervals of AF are irregular, the multiplication of SD1 (standard deviation of long axis) and SD2 (standard deviation of short axis) is used as the third feature in the poincaré plot. Additionally, irregular RR intervals reduce the probability of the highest bin (PHB) in the RR histogram. The PHB can be calculated as the reciprocal of the HRV triangular index (HTI). To simplify the calculations, the complement of 1 for the PHB (CPHB) is obtained as (1–PHB) and is used as the fourth feature. The fifth feature is the high-frequency amplitude of the fibrillatory wave. AF has a strong high-frequency component, because fibrillatory waves are normally included in the non-QRS complex of AF. The high-frequency component of the fibrillatory wave can be detected using Short-time Fourier Transform. In the spectrogram of Figure 7, the horizontal and vertical axes represent the time and frequency, respectively, and the pseudo-color represents the power. If the fibrillatory wave is severe, the power value increases. Lower values of the AA and PHB and higher values of the BPM, the multiplication of the two standard deviations (MSD), and the frequency of the fibrillatory wave correspond to more severe AF.

According to the foregoing explanation, the five normalized features for AF detection are calculated as follows: (7)$$\left\{ \begin{array}{c} \mathrm{AA}=(PA_{\max}-PA_{\mathrm{avg}})/(PA_{\max}-PA_{\min)}\\ \mathrm{HR}=({\mathrm{BPM}}_{\mathrm{avg}}-{\mathrm{BPM}}_{\min})/({\mathrm{BPM}}_{\max}-{\mathrm{BPM}}_{\min})\\ \mathrm{SD}=({\mathrm{MSD}}_{\mathrm{avg}}-{\mathrm{MSD}}_{\min})/({\mathrm{MSD}}_{\max}-{\mathrm{MSD}}_{\min})\\ \mathrm{CPHB}=({\mathrm{CPHB}}_{\mathrm{avg}}-{\mathrm{CPHB}}_{\min})/({\mathrm{CPHB}}_{\max}-{\mathrm{CPHB}}_{\min})\\ \mathrm{OSC}=({\mathrm{OSC}}_{\mathrm{avg}}-{\mathrm{OSC}}_{\min})/({\mathrm{OSC}}_{\max}-{\mathrm{OSC}}_{\min}), \end{array}\right.$$
where OSC represents the average power of the spectrogram, and the subscript ‘avg’ denotes the average of each feature obtained over a given time period. The minimum and maximum values for normalizing each feature are as follows: ${\mathrm{PA}}_{\max}=\beta^{-1}\times \left(3\alpha \times F\right)\left(2\alpha \times F\right)/2$, β=10, α(=0.4), ${\mathrm{PA}}_{\min}=0$, ${\mathrm{BPM}}_{\max}=100$, ${\mathrm{BPM}}_{\min}=0$, ${\mathrm{MSD}}_{\max}=$0.01, ${\mathrm{MSD}}_{\min}=0$, ${\mathrm{CPHB}}_{\max}=1$, ${\mathrm{CPHB}}_{\min}$ = 0, ${\mathrm{OSC}}_{\max}=15$, and ${\mathrm{OSC}}_{\min}=0$. α represents the time (in seconds) of one grid in the ECG graph paper. In ${\mathrm{PA}}_{\max}$, the first and second parentheses indicate the base and height of a typical P wave, respectively [48]. Half of the product of these values corresponds to the area of the P wave. β represents the ratio control factor of the PA. The max of the BPM is set based on tachycardia, which means more than 100 beats per minute. One standard deviation value of 0.1 was determined assuming that the difference between the current and next RR intervals for a stable heart rate was 0.1 or less. As a result, the max of the MSD was set at 0.01. Since the PHB is a probability between 0 and 1, the CPHB also has a value in the same range. And the fibrillatory waves have the atrial rate from 300 to 600 waves per minute with varying morphology and high frequency [49]. The limit of the high waves of the fibrillation wave is occasionally set up to 650 [50]. Taking into account the waves per minute of the fibrillation wave, the min and max values of the OSC were set.

## 3. Results

### 3.1. R-Peak Detection Performance Using MITDB

Figure 9 shows the R-point detection result, difference, max filter, and adaptive threshold for various MITDB records. R peaks were detected normally for the normal sinus rhythm (N) of Record 100. Certain parts of Record 104 consisted of Paced (/) and a fusion of paced and normal (f). The “ ‘/’ rhythm exhibited a wide and high T wave, and the ‘f’ rhythm exhibited a spiked R peak. Examining the difference between the original signal and the filtered signal reveals that the S and T waves of the ” ‘/’ rhythm and the additional R wave of the ‘f’ rhythm were suppressed, while the R peak was improved. An isolated QRS-like artifact (|) occurred between the 9th and 10th normal sinus rhythms of Record 105, but the proposed method successfully suppressed the artifact. Additionally, Record 108 contained baseline wander (motion artifact), but R peaks were detected normally. The experimental results confirmed that the proposed method is robust to artifacts such as deformation of S and T waves and baseline wander.

To evaluate the performance of the detection algorithm, several indices were introduced, including the number of true positives (TP, QRS complexes detected as QRS complexes), the number of false negatives (FN, QRS complexes not detected as QRS complexes, number of incorrectly rejected QRS complexes), number of false positives (FP, non-QRS complexes detected as QRS complexes, number of incorrect QRS predictions). The sensitivity (Se = TP/(TP + FN)) is the percentage of true beats that were correctly detected by the algorithm. The positive prediction (+P = TP/(TP + FP)) is the percentage of beat detections that were true beats. The detection error (DER = (FP + FN)/(TP + FN)) is the ratio of the number of false detections to the total number of detected heartbeats. The detection performance for MITDB is shown in Table 2. An average QRS detection rate of 99.62%, a sensitivity of 99.82% and a positive prediction of 99.80% are obtained against all recordings of MITDB. 

Record 105 is a tape that makes it difficult to detect the R peak owing to heavy noise and is widely used for comparison and verification of algorithms. Table 3 presents a comparison of the existing algorithms and the proposed method using Record 105. The proposed method ranks second among the published results of other algorithms, confirming its excellent performance.

The time cost of the proposed method was analyzed for several records from the MITDB (360 Hz) using MATLAB R2020B in Windows 10 with an Intel i9 4-core central processing unit and 128 GB. Figure 10 shows the calculation time per 1000 iterations over 11 s of Records 100, 104, 105, and 108. The respective average processing time was 22.36 ms for Record 100, 22.05 ms for Record 104, 22.36 ms for Record 105, and 21.71 ms for Record 108. The total processing average time for an 11-s period was approximately 22.12 ms, indicating that the proposed method is suitable for real-time ECG signal processing.

### 3.2. Arrhythmic Feature Detection Using Various ECG Data

Various ECG data were used to detect arrhythmic features. Bigeminy is the signal measured by VP-100, while trigeminy, VT, and SVT were generated by the MS 400 simulator. Additionally, Record 114 containing PVC was used for the PVC detection experiment. Figure 11 shows the R-peak detection results for these arrhythmias. In the case of Bigeminy, RR intervals of 0.6 s and approximately 0.9 s were repeated, whereas in trigeminy, RR intervals of 0.55, 0.78, and 0.90 s were repeated. Bigeminy and trigeminy contain PVCs, which have very low negative R peaks.

The normal R peak in these signals is difficult to detect because it has a smaller difference value than the PVC in the difference signal. Nevertheless, the normal R is also detected owing to the lower R-peak detection threshold via the adaptive threshold method of the proposed technique. The normal R of Record 114 containing PVCs was not difficult to detect, in contrast to the cases of bigeminy and trigeminy. In VT and SVT, the R-peak detection threshold was increased owing to the high HR, and the R peak was normally detected. The merged P-T waves in SVT were normally detected, whereas fragmented P-T waves were not detected.

Table 4 shows the performance evaluation of the arrhythmias tested in Figure 11. The evaluation of MITDB 114 was excluded because it is already indicated in Table 2. There was no false R detection for all arrhythmias (FP = 0). While there were R peaks that were not detected in bigeminy and trigeminy, the R peaks of VT and SVT were almost completely detected. This is due to the non-detection of the weakly negative R peak at bigeminy and the lower two R peaks at trigeminy in one sliding window. It can be seen that the R peaks of bigeminy and trigeminy are not easy to detect completely due to R peaks of various heights. 

### 3.3. HRV Analysis of Various ECG Signals

The HRV can be measured via time- and frequency-domain methods for evaluating the sympathovagal balance [51]. Table 5 presents the variables related to time- and frequency-domain HRV measurements. The HRV is measured by the variation in the RR interval. This variation is controlled by a primitive part of the nervous system called the autonomic nervous system [52]. RMSSD, NN50, and pNN50 represent the short-term cardiac variability, indicate parasympathetic nerve activity, and are closely associated with sudden death and AF in epilepsy [53]. In frequency analysis, the low- and high-frequency components represent the activity of the sympathetic and parasympathetic nerves, respectively. Low sympathetic nerve activity indicates increases in the blood pressure and HR, and low parasympathetic activity indicates reductions in the blood pressure and HR.

The HRV results for MITDB and various arrhythmias are presented in Figure 12 and Table 6 The ECG signals used in the experiment were MITDB (each 10 min long), bigeminy (approximately 56 s), trigeminy (approximately 56 s), PVC (9 min 30 s, MITDB 114), VT (1 min), and SVT (1 min) signals. In Table 4, the bold blue numbers indicate abnormal values that must be examined carefully. Compared with arrhythmias, the records used in the MITDB exhibited relatively normal behavior in HRV visualization. For example, in the case of Record 100, owing to the constant RR intervals, points corresponding to the RR intervals were concentrated in one place in the histogram and poincaré plot. However, overall, Record 104 with partial tachycardia and Record 108 with bradycardia exhibited higher RMSSD, NN50, and pNN50 values than the other MITDB records used in the experiment. Record 108 exhibited the highest HTI and TINN among the MITDB records, indicating that the RR intervals were highly irregular compared with the other signals.

In contrast to the aforementioned MITDB, the arrhythmias exhibited abnormal distribution characteristics in the HRV visualization. First, in bigeminy, a short RR interval of 0.6 s and a long RR interval of 0.9 s (on average) were repeated. In the RR interval trend, the short periods were constant, and the long periods changed irregularly. In trigeminy, a short period of 0.55 s, an intermediate period of 0.78 s, and a long period of 0.90 s were repeated. For bigeminy and trigeminy, there were two or three independent groups of RR interval points on the histogram and poincaré plot, owing to their periodicity. MITDB Record 114 was a bradycardia signal with 47 PVCs out of a total of 1879 beats, and these PVCs were responsible for scattered RR interval points on the histogram and poincaré plot [54]. Because the VT and SVT signals were generated by simulators with very short periods, they yielded very small RR intervals that were concentrated at one point on the poincaré plot. In particular, SVT exhibited a very high mean HR. Among the arrhythmias, bigeminy and trigeminy exhibited the highest pNN50. Record 114 exhibited the highest NN50 but a low pNN50, because the number of NN50 was low compared with the overall signal length. 

On the PSD, arrhythmias excluding Record 114 exhibited a higher HF power and very low LF and HF percentages compared with the other MITDB records. This indicates that these arrhythmias reflect very low activity of the sympathetic and parasympathetic nerves. The LF/HF ratios for Trigemini, PVC, and VT were <0.4, indicating that the degree of balance of autonomic nerves is low.

### 3.4. AF Detection

Excluding records with problems with data length and data loading, the partial records of AFDB were used to extract the AF features. In these signals, the normal beat period and the AF period were alternately repeated. In the AF signal used in the experiment, a length of 24 min after 46 s was used, and Figure 13 shows the beginning 12 s of these AF signals. Figure 14 presents the normalized feature values of the AF signals in polar coordinates. Figure 14 shows the visualization of VT and SVT using AF detection technology. It can be seen that arrhythmias other than AF do not show a large area in AF visualization. Also, the normal signal will have a very small area in its visualization. As shown, more severe AF corresponded to larger AF feature values in the polar coordinates. The severity of the AF is visually displayed. Records 04126 and 06995 exhibited better HRs. Record 06426 and 07161 show the concentrations of the RR intervals in the poincaré plot than the other signals. Record 06426 and 08378 were considered to be the most severe AF visually. For Records 07162, the high-frequency component in the spectrogram was small. Record 08405 exhibited better AA than the other signals.

Table 7 presents the five normalized feature values for the AF signals. Among the AF signals used in the experiment, Record 04043 for HR, Record 04746, 06995, 07162, 07910, and 08215 for AA, Record 06426 and 07162 for CPHB and SD, and Record 06426 and 08378 for OSC exhibited the highest (worst) feature values. An AF analysis revealed that the AF severity was in the following order: Records 06426, 08378, 06995, 05121, 04126, 04043, 07162, 05261, 08405, 05091, 07910, 07879, 04746, and 08215. The test AF signals used in the experiment show high average values from 0.61 to 0.91 in normalized AF features. In addition, each normalized AF feature value for test AF signals shows a range of 0.63 to 0.94. It can be inferred that the proposed features can be used for AF detection through these experiments.

## 4. Discussion

Recently, ECG measuring devices having various sampling rates from smart bands to 12-channel ECG measuring devices have been developed. One type of ECG measuring device is connected to a database server of a hospital for storing and analyzing biological signals. However, heart-disease patients may use different measuring devices according to their symptoms, disease names, and measurement periods. Additionally, various ECG compression techniques [55,56,57] are being developed to increase the transmission efficiency of the measured ECG signals. During the compression process, the sampling rate of the original ECG signal may change. In this ECG measurement and transmission environment, the ECG processing and analysis algorithm must respond to various sampling rates, for which the proposed method offers a solution.

The proposed R detection and HRV analysis technology can be directly utilized for real-time heart rate monitoring through a wearable ECG measuring device [58]. In addition, HRV parameters related to autonomic and non-autonomous nervous systems can be used as additional parameters for blood pressure estimation using photoplethysmography [59].

Previous arrhythmia studies have been limited to the field of detection, and HRV analysis for arrhythmia has been insufficient. The HRV analysis results for the arrhythmias discussed in this paper can be used to increase the accuracy of arrhythmia detection. Currently, the HRV is analyzed in the time and frequency domains according to the time difference in the time domain of the detected R peak. However, the height information of the R peak is also an important parameter. Therefore, research on a new HRV based on the height difference in the range domain of the detected R peak must be performed.

In this study, AF was visualized using polar coordinates. The visualization of major arrhythmias can be useful for the early detection of arrhythmias and is an efficient method for analyzing the heart conditions of heart-disease patients. According to the results of present study, other major arrhythmias can also be visualized using their own features. Visualization studies involving arrhythmia can significantly affect the development of applications for heart-condition analysis of arrhythmia patients.

## 5. Conclusions

This paper proposes an R-peak detection method for processing ECG signals measured by ECG measuring devices with different sampling rates. Additionally, a method for detecting major arrhythmia was presented, and an HRV analysis was performed. A visualization method for AF was suggested according to its characteristics. In the future, various measurement devices may be connected to one hospital database server for efficiently measuring biometric signals. In this case, the proposed method can be universally employed regardless of the connected measuring device. Furthermore, the visualization of arrhythmia will be accelerated in the development of applications for heart-disease patients in the future.

## Figures and Tables

**Figure 1 sensors-20-06144-f001:**
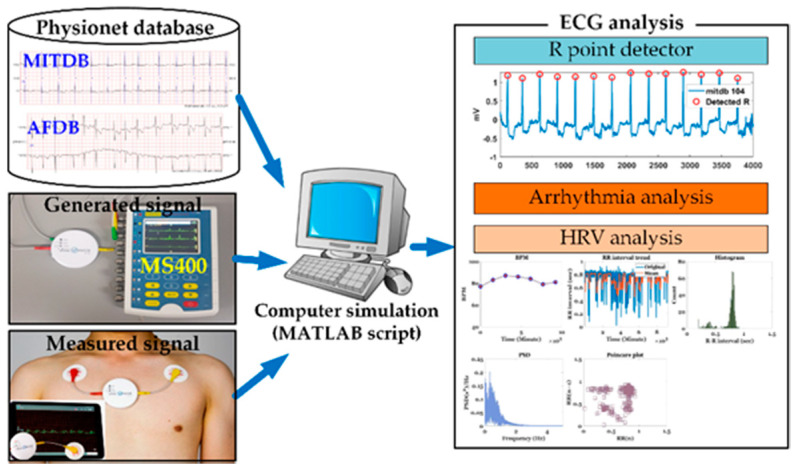
Simulation environment.

**Figure 2 sensors-20-06144-f002:**
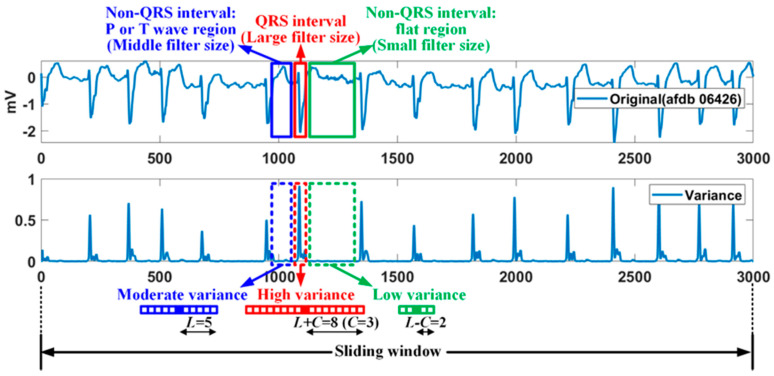
Relationship of variance and filter size: an example of 250 samples/s.

**Figure 3 sensors-20-06144-f003:**
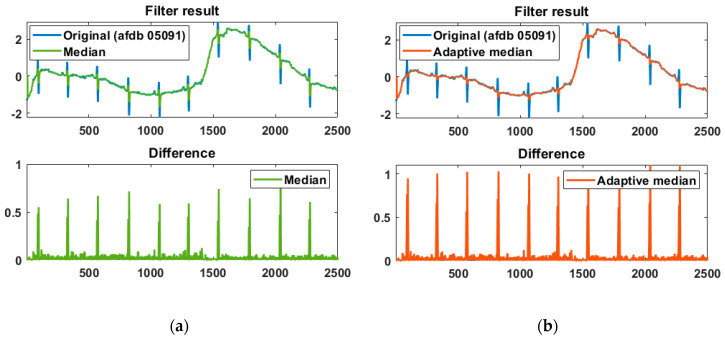
Difference comparison of the original signal and the filter result using (**a**) the normal median filter and (**b**) the adaptive median filter.

**Figure 4 sensors-20-06144-f004:**
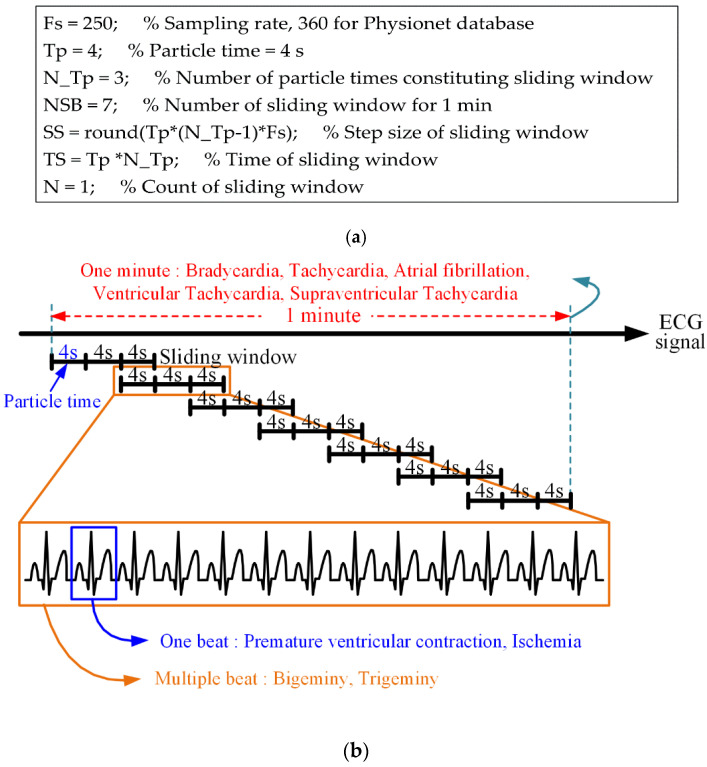
Sampling rate-based sliding-window structure; (**a**) variable configuration of the sliding window based on the sampling rate and (**b**) detection structure for various arrhythmias.

**Figure 5 sensors-20-06144-f005:**
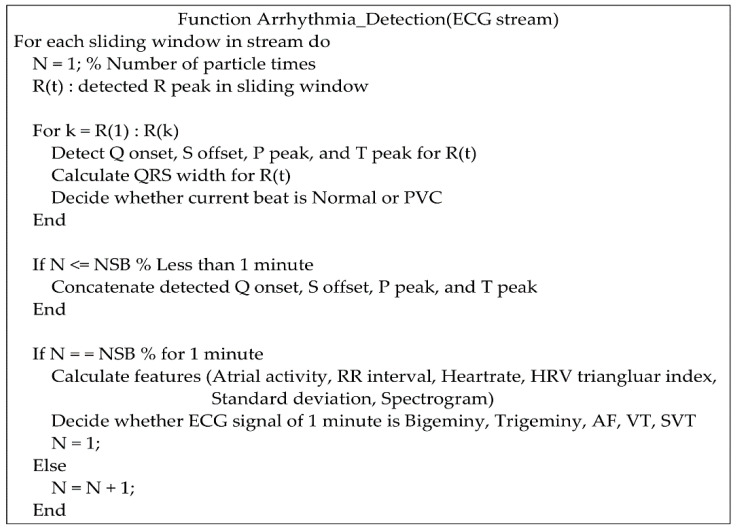
Pseudocode for the detection of various arrhythmias.

**Figure 6 sensors-20-06144-f006:**
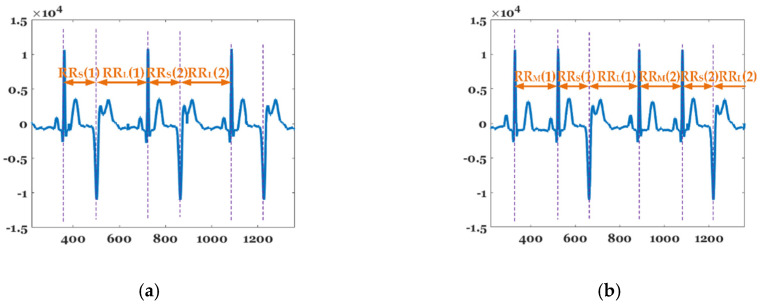
Features of (**a**) bigeminy and (**b**) trigeminy.

**Figure 7 sensors-20-06144-f007:**
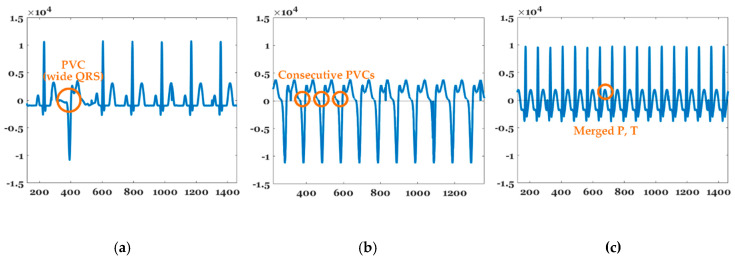
Features of (**a**) PVC, (**b**) VT, and (**c**) SVT.

**Figure 8 sensors-20-06144-f008:**
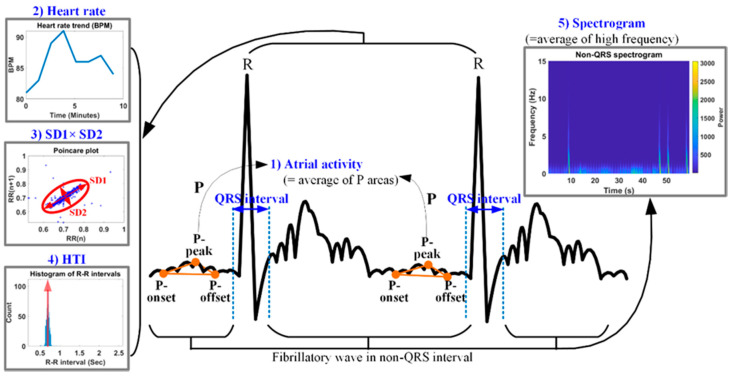
Five features for AF detection.

**Figure 9 sensors-20-06144-f009:**
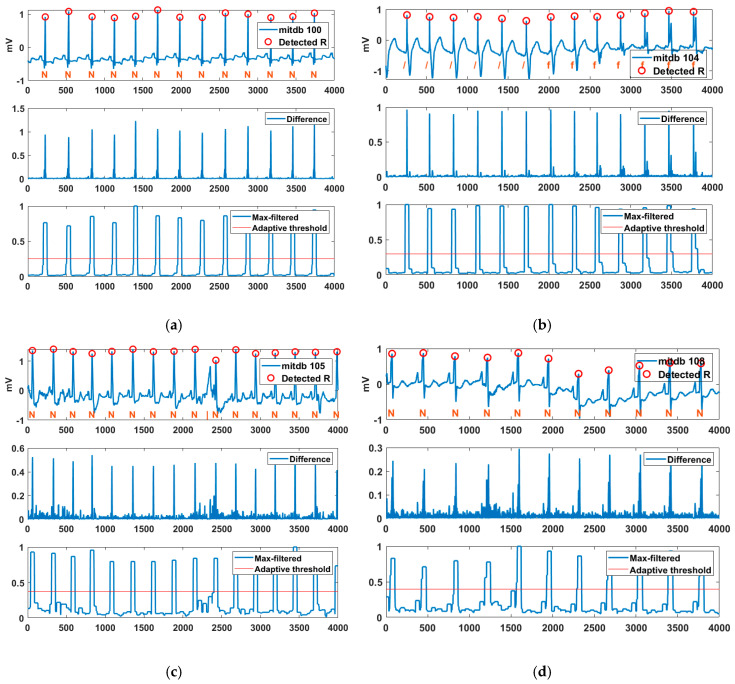
Detected R peaks for MITDB Records (**a**) 100, (**b**) 104, (**c**) 105, and (**d**) 108.

**Figure 10 sensors-20-06144-f010:**
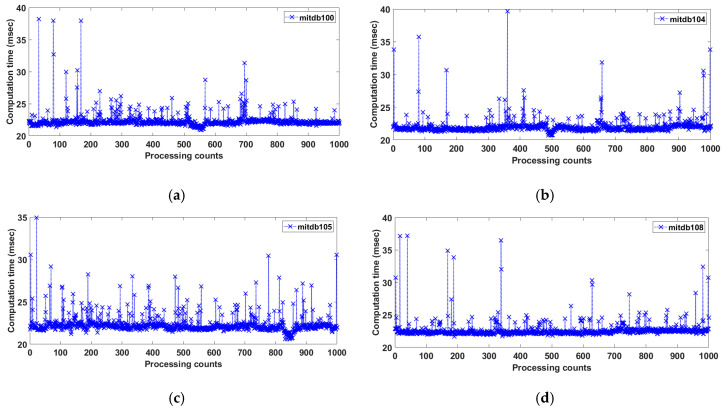
Computation time per 1000 iterations for 11 s of Records (**a**) 100N112, (**b**) 104/328, (**c**) 105N364, and (**d**) 108N270.

**Figure 11 sensors-20-06144-f011:**
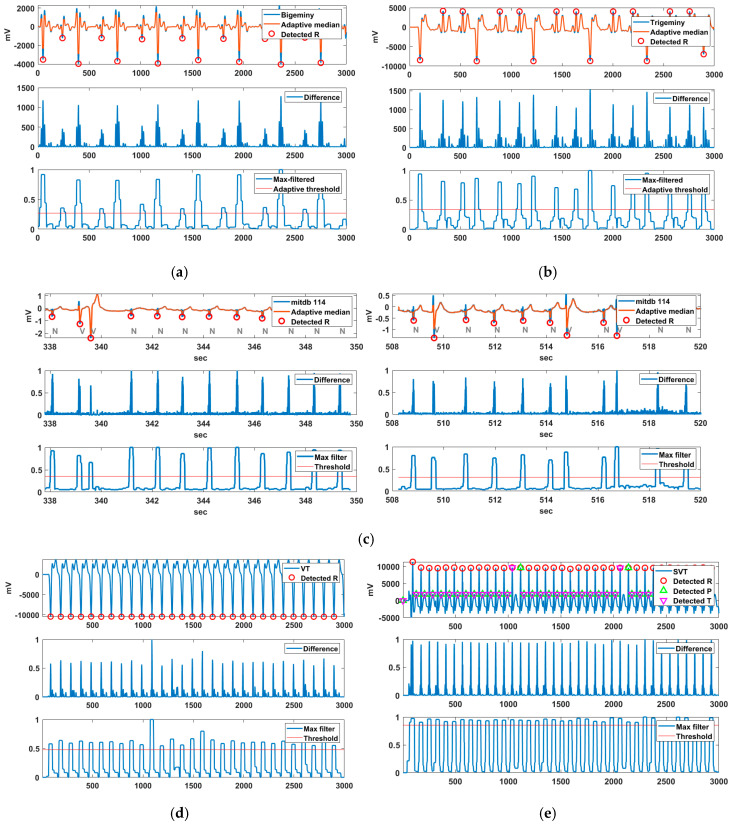
R-peak detection results for (**a**) bigeminy, (**b**) trigeminy, (**c**) MITDB 114 including PVC, (**d**) VT, and (**e**) SVT.

**Figure 12 sensors-20-06144-f012:**
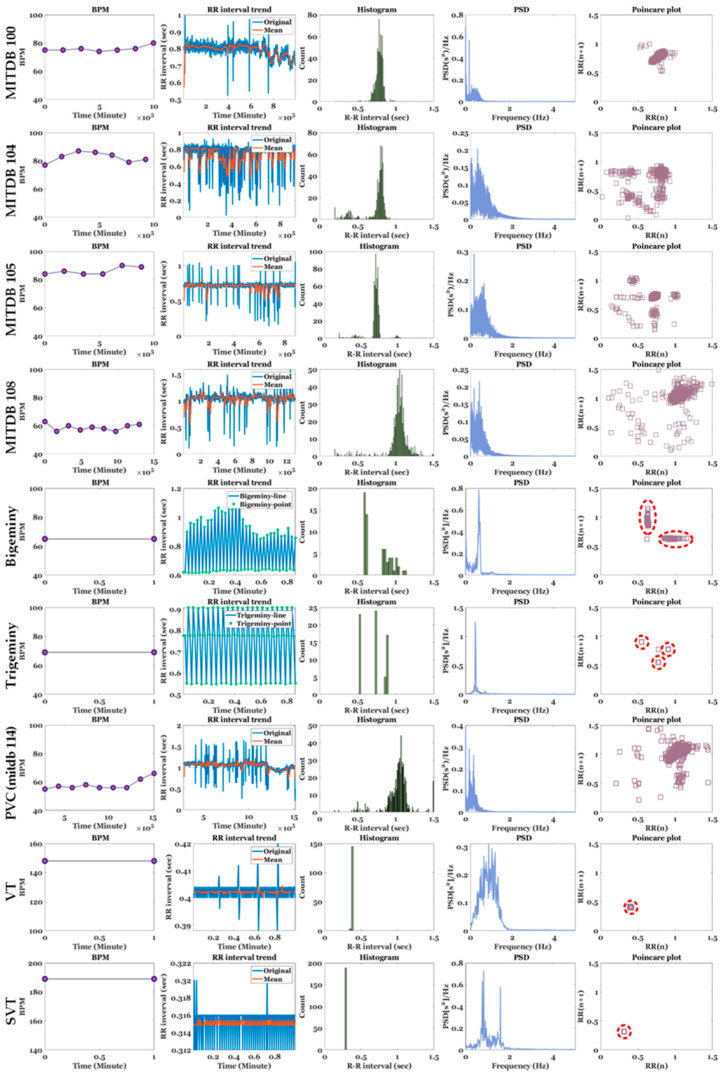
Comparison of BPM, RR interval, RR histogram, PSD, and poincaré plot for MITDB Records (100, 104, 105, and 108), bigeminy, trigeminy, mitdb 114 including PVC, VT, and SVT.

**Figure 13 sensors-20-06144-f013:**
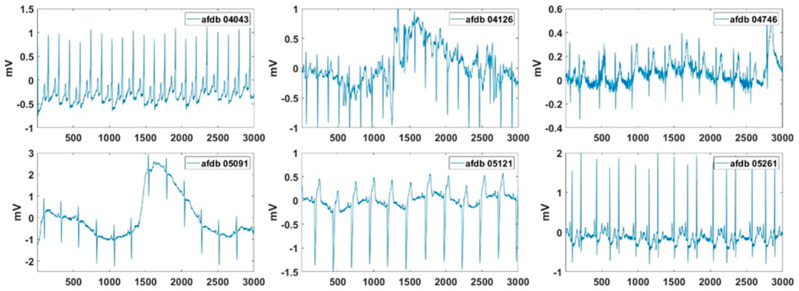
Test signals of AFDB used for AF detection.

**Figure 14 sensors-20-06144-f014:**
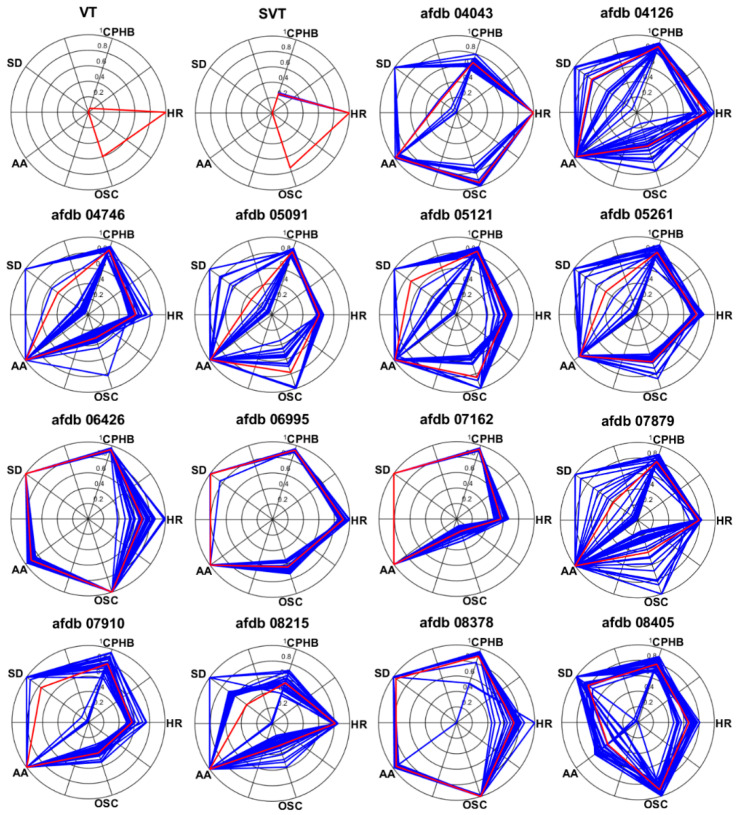
Visualization of test AF signals in in AFDB.

**Table 1 sensors-20-06144-t001:** Processing unit for various arrhythmias.

Processing Unit (per)	Kinds of Arrhythmia
One beat	Premature ventricular contraction (PVC)
Multiple beat (or one minute)	Bigeminy, trigeminy
At least one minute	Bradycardia, tachycardia, ventricular tachycardia (VT), supraventricular tachycardia (SVT), atrial fibrillation (AF)

**Table 2 sensors-20-06144-t002:** Performance evaluation of the proposed method using the MITDB.

Tape	Total	FN	FP	Se [%]	+P [%]	DER [%]	Tape	Total	FN	FP	Se [%]	+P [%]	DER [%]
100	2273	0	0	100	100	0	201	1963	7	3	99.64	99.85	0.51
101	1865	2	5	99.89	99.73	0.38	202	2136	4	5	99.81	99.77	0.42
102	2187	1	1	99.95	99.95	0.09	203	2980	18	21	99.40	99.30	1.31
103	2084	0	1	100	99.95	0.05	205	2656	7	3	99.74	99.89	0.38
104	2229	3	12	99.87	99.46	0.67	207	1862	9	7	99.52	99.62	0.86
105	2572	11	18	99.57	99.30	1.13	208	2955	13	5	99.56	99.83	0.61
106	2027	5	6	99.75	99.70	0.54	209	3004	4	5	99.87	99.83	0.30
107	2137	2	5	99.91	99.77	0.33	210	2650	14	6	99.47	99.77	0.75
108	1774	7	27	99.61	98.49	1.92	212	2748	3	5	99.89	99.82	0.29
109	2532	3	4	99.88	99.84	0.28	213	3251	2	5	99.94	99.85	0.22
111	2124	4	3	99.81	99.86	0.33	214	2265	2	2	99.91	99.91	0.18
112	2539	0	1	100	99.96	0.04	215	3363	1	3	99.97	99.91	0.12
113	1795	2	4	99.89	99.78	0.33	217	2209	5	2	99.77	99.91	0.32
114	1879	2	5	99.89	99.73	0.37	219	2154	2	5	99.91	99.77	0.32
115	1953	0	1	100	99.95	0.05	220	2048	1	1	99.95	99.95	0.10
116	2412	9	5	99.63	99.79	0.58	221	2427	5	2	99.79	99.92	0.29
117	1535	2	3	99.87	99.80	0.33	222	2483	7	2	99.72	99.92	0.36
118	2278	0	1	100	99.96	0.04	223	2605	5	1	99.81	99.96	0.23
119	1987	1	3	99.95	99.85	0.20	228	2053	6	8	99.71	99.61	0.68
121	1863	4	2	99.79	99.89	0.32	230	2256	3	2	99.87	99.91	0.22
122	2476	0	1	100	99.96	0.04	231	1571	1	2	99.94	99.87	0.19
123	1518	3	1	99.80	99.93	0.26	232	1780	3	4	99.83	99.78	0.39
124	1619	2	1	99.88	99.94	0.19	233	3079	5	1	99.84	99.97	0.19
200	2601	1	7	99.96	99.73	0.31	234	2753	3	2	99.89	99.93	0.18
							Total	109510	194	219	99.82	99.80	0.38

**Table 3 sensors-20-06144-t003:** Performance comparison with other algorithms using Record 105.

Method	FP	FN	DER (%)	Ref
Linear adaptive filter	40	22	2.41	[12]
Bandpass filter	67	22	3.46	[10]
Generic algorithm	86	5	3.54	[16]
Wavelet transform	31	13	1.17	[17]
Wavelet denoising	5	78	3	[18]
3M method	19	7	1.01	[19]
Derivative-max filter	20	13	1.28	[20]
Proposed method	18	11	1.13	

**Table 4 sensors-20-06144-t004:** Performance evaluation of test arrhythmias.

Arrhythmias	Length	No. of Beats	FN	FP	Se(%)	+P(%)	DER(%)
Bigeminy	56 sec	72	7	0	90.28	100	9.72
Trigeminy	56 sec	79	10	0	87.34	100	12.66
VT	59 sec	149	1	0	99.33	100	0.67
SVT	59 sec	191	2	0	98.95	100	1.05

**Table 5 sensors-20-06144-t005:** Variables related to time- and frequency domain- HRV measurements.

	Variables	Units	Meaning
Time domain	Mean HR	[s]	-
	HR std.	[s]	-
	RR mean	[s]	Average RR interval in the window of measurement
	NN50	[count]	Number of adjacent RR intervals that varied by more than 50 ms
	pNN50	[%]	Percentage of adjacent RR intervals that varied by more than 50 ms
	RMSSD	[s]	Root mean square of difference between coupling intervals of adjacent RR intervals
	HTI	-	Reciprocal of probability of the highest bin of histogram of RR intervals
	TINN	-	The baseline width of the distribution measured as a base of a triangle
Frequency domain	VLF power	[ms2]	Power from very low frequency (0 Hz~0.04 Hz)
	LF power	[ms2]	Power from low frequency (0.04 Hz~0.15 Hz)
	HF power	[ms2]	Power from high frequency (0.15 Hz~0.40 Hz)
	VLF	[%]	(VLF Power/Total Power) × 100
	LF	[%]	(LF Power/Total Power) × 100
	HF	[%]	(HF Power/Total Power) × 100
	LF/HF	-	Sympathovagal balance

**Table 6 sensors-20-06144-t006:** HRV results for MITDB records and various arrhythmias.

HRV	MITDB				Arrhythmias				
	100	104	105	108	Bigeminy	Trigeminy (gen.*)	PVC (mitdb114)	VT (gen. *)	SVT (gen. *)
RR mean	0.79	0.73	0.70	1.03	0.79	0.74	1.04	0.40	0.32
RR std.	0.05	0.17	0.11	0.17	0.17	0.15	0.18	-	-
Heart rate mean	75.86	82.43	86.17	58.89	65	69	58	148	189
Heart rate std.	1.95	3.65	2.71	2.37	-	-	3.64	-	-
RMSSD	0.05	0.19	0.15	0.20	0.33	0.25	0.25	0.01	0.00
NN50	23	118	49	142	31	45	144	-	-
pNN50	3.96	20.31	8.43	24.44	48.44	66.18	26.23	-	-
HTI	7.66	8.56	6	11.64	3.42	2.88	12.50	1.02	1
TINN	0.21	0.25	0.11	0.39	0.06	0.03	0.31	0.06	0.03
LF power	22.91	29.33	25.08	32.41	37.12	10.39	16.25	15.43	27.34
HF power	77.09	70.67	74.92	67.59	62.88	89.61	83.73	84.57	72.66
LF	2.93	5.18	4.12	5.34	0.61	0.08	2.91	0.25	0.20
HF	9.84	12.48	12.29	11.13	1.03	0.66	14.98	1.35	0.52
LF/HF	0.30	0.42	0.33	0.48	0.59	0.12	0.19	0.18	0.38

gen. *: generated.

**Table 7 sensors-20-06144-t007:** Five normalized feature values for various AF signals in AFDB.

Records	HR	CPHB	SD	AA	OSC	Average
04043	1	0.68	0.26	0.98	0.96	0.78
04126	0.90	0.90	0.73	0.97	0.46	0.79
04746	0.61	0.88	0.48	1	0.32	0.66
05091	0.61	0.85	0.33	0.99	0.78	0.71
05121	0.64	0.86	0.73	0.99	0.85	0.81
05261	0.77	0.84	0.50	0.91	0.64	0.73
06426	0.70	0.95	1	0.91	1	0.91
06995	0.92	0.94	0.99	1	0.64	0.90
07162	0.57	0.95	1	1	0.17	0.74
07879	0.79	0.79	0.34	0.99	0.43	0.67
07910	0.57	0.80	0.77	1	0.42	0.71
08215	0.82	0.55	0.41	1	0.30	0.61
08378	0.74	0.90	0.96	0.96	1	0.91
08405	0.67	0.79	0.81	0.49	0.92	0.73
Average	0.74	0.83	0.67	0.94	0.63

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
