# Peer review of "An Adaptive Median Filter Based on Sampling Rate for R-Peak Detection and Major-Arrhythmia Analysis"

_sensors, 2020, doi:10.3390/s20216144_

Round 1

Reviewer 1 Report

In the submitted paper "Adaptive Median Filter Based on Sampling Rate for R peak detection and Major Arrhythmia Analysis" authors proposed a new algorithm for R peak detection that is independent on the sampling rate of the measurement and new algorithms for the detection of important arrhythmias. 

Comments:

Chapter 1. Introduction

Minor comments:

Lines 48 - 53 contain a description of why it is necessary to propose an algorithm for ECG processing independent on the sampling rate. However, the paragraph is the just description that there is some ratio between the sampling rate and reachable quality of processing and speed of calculation. I think it would be better to mention that there are several different reasons to measure ECG. There are telemedicine purposes, localization heart attack purpose, cardiac resynchronization therapy purposes, and so on. Furthermore, these are the reasons why we need more than one sampling rate.

Chapter 2.2. R peak detection

Minor comments:

There are equations for the median filter (1) and adaptive filter size determination rules (2). It should be declared an equation for the difference signal as well. Of course, it is clear to see it from Figure 2, but the equation would give a complete definition of the entire process.

Major comments:

From my perspective, the most important comment to this part is

the question, what is the background for the build of rules for defining the size of the adaptive filter? There are no references for previous works, and there is no description of how authors created the rules. Thus, it looks like magic. Why the authors use constants one third and two thirds? Moreover, the description of the rules it seems to me very brief. It would be suitable to consider depicted the process in the figure.

Chapter 2.3. Detection of Important Arrhythmias

Minor comments:

Line 106: However, Bigeminy ... It should be small "b".

Line 107: in addition, ... It should be capital "I".

Figure 3a: The description of TS is not right. TS is not the time, because there is multiplying of the Particle time and Sampling rate (there is no unit). The number gives the count of samples in the Sliding window. 

Figure 4: There should be mentioned initialization of N (N=1) at the start of the algorithm.

Major comments:

There is the same comment as in the previous chapter 2.2. There is no defined reason why the Particle time is 4s, the step size of 8s, the overlap is 4s and why there are 3 Particle times in the Sliding window. It should be mentioned why we cannot use 2s Particle time, 4s Sliding window or something else. If it is an empirical result from experiments, there should be some text about the optimization of these parameters. In this form, it looks like information from manual not from a scientific paper.

Figure 4: If I understand right, the arrhythmia detection process defines the Sliding window, and within each Sliding window, the R peak detection algorithm defines specific filter windows with different sizes. In other words, two processes use different windows in different time of processing and always the filter windows are defined within the currently used Sliding window.  Is my understanding correct?

Chapter 2.3.2 Bigeminy and trigeminy

Minor comments:

Lines 137 and 138: ... represent the average length of the short, middle and long period in the observation time.

Major comments:

Lines 144 and 145: Lines 144 and 145: Why the DiffLS and DiffMS are determined to 10? What was the process behind the determination of these values?

Lines 149 and 150: The sentence does not describe the rule for a trigeminy recognition defined in equation 4.

Based on Figure 5: I will omit the question of what R peak is, whether R peak can be negative. I am not a physician, but I know that this question is no easy. In the depicted PVC beat cases, the negative peak can be probably rS or QS. However, it is just the question of the terminology.  Let say that there is something like reverse R peak or mirrored peak. My question is how the algorithm can detect the negative peak. It means that there is absolute value in the calculation the difference signal. It is the next reason why to define the difference signal calculation by the equation. 

Chapter 2.3.3. PVC, VT, and SVT

Minor comments:

Figure 6: The legends are not exactly correct. In all figures, the signal is ECG not PVC, VT or SVT.

Chapter 2.3.4. AF

Minor comments:

Line 168: ... less than 120 ms wide ...

Lines 185 and 186: "... component of the fibrillatory wave can be detected using spectrogram or Short-time Fourier Transform." However, the spectrogram is just the visualization of the transformation from the time domain into the time-frequency domain, such as the Short-time Fourier Transform. Authors should reformulate this sentence.

Major comments:

Based on Figure 7: The spectrogram in the left part of the figure has defined frequency axis (y-axis) from 0 up to 5 Hz. It is a visualization of the high-frequency component of fibrillatory waves. However, the fibrillatory waves have the atrial rate from 300 to 600 waves per minute. In simple calculation (atrial rate/60), we obtain frequency 5 to 10 Hz. So, the picture is misleading. However, this is not my main point. Because, authors defined the normalizing OSC values correctly in the settings of the calculation (OSCmax = 15, OSCmin = 0), but there is again no explanation of how the values were acquired. It is same for other normalizing values in the AF features calculation process.

Chapter 3. Results

The main problem with results represents not always a clear and reliable selection of tested signals.

Chapter 3.1.

Major comments:

The R peak detection was detected only on MITDB and only on 20 signals from the MITDB. The reason why the authors use only the part of one database is not declared.  It is not necessary to show all results for all databases. However, it would be suitable to show at least the average results and not only for selected records but also for all record in the database. This selection of records leads to thinking that the authors selected only those records where the results were applicable for the paper. It is simply not an acceptable procedure. Authors declared in the chapter that they used Record 105 for comparison with other works because it is used in the larger number of works. This part is ok. However, it is not the reason why not show and comment on the results of R peak detection on all databases and all records.

Chapter 3.2. Arrhythmia feature detection using various ECG data

Major comments:

There are just several pictures with some results, but there is no summary of results in this case. In fact, record 114 from the MITDB has the result of detection in the previous chapter. However, the rest of the records is without detection evaluation. They are simulated signals and there is not clear an obstacle to obtaining the detection evaluation.  Moreover, there is not clear how many beats/signals were simulated and how many differences the signals contained (how was different variability of signals).

Chapter 3.2. HRV analysis of various ECG signals

Minor comments:

The number of the chapter is wrong. It should be chapter 3.3.

Figure 11: There is no a,b,c,d,e marking in the figure.

Table 4: There is not clear why some values are bold.  Is it statistical significance? There is no mentioned test in the text.

Major comments:

In fact, this chapter is a case study because there is used only one record for each type of signal (there is not distinguished that signals are simulated). Therefore, the results cannot be generalized. Table 4 and Figure 11 are just examples of how the HRV analysis parameters and the five features for AF detection can be distributed in specific cases of the signal. However, the generalization of the approach is zero because there is no statistical evaluation of the parameters and features/classification. It is the first step in the using of HRV for detection of other arrhythmias than only AF.  However, without evaluation, it is not so strong how it should be. Finally, there is not clear how authors intend the combination of HRV parameters and defined five features.

Chapter 3.3. AF detection

Minor comments:

The number of the chapter is wrong. It should be chapter 3.4.

Line 308: There is no mentioned record 05121 in the selection of records from AFDB.

Table 5: There is not clear why different colors highlight several values.

Major comments:

There is no clear why authors selected only six records of AFDB from 25 total.

The problem is the same as in the previous chapter. There are observable differences between the features' representations. The visualization of arrhythmias seems to be promising. However, there are given results of specific signals always only from one subject.

Without statistical evaluation, there is impossible to evaluate the influence of the variability among records from different patients.

Author Response

# Reviewer 1

Comments and Suggestions for Authors

In the submitted paper "Adaptive Median Filter Based on Sampling Rate for R peak detection and Major Arrhythmia Analysis" authors proposed a new algorithm for R peak detection that is independent on the sampling rate of the measurement and new algorithms for the detection of important arrhythmias. 

Ans) The whole manuscript was edited by native speaker.

Comments:

Chapter 1. Introduction

Minor comments:

Lines 48 - 53 contain a description of why it is necessary to propose an algorithm for ECG processing independent on the sampling rate. However, the paragraph is the just description that there is some ratio between the sampling rate and reachable quality of processing and speed of calculation. I think it would be better to mention that there are several different reasons to measure ECG. There are telemedicine purposes, localization heart attack purpose, cardiac resynchronization therapy purposes, and so on. Furthermore, these are the reasons why we need more than one sampling rate.

Ans) Based on your advice, the following has been added.

(Line 49) Typically, ECGs are used for arrhythmia detection and prediction or in emergency care situations. Recently, ECG measuring devices and telemedicine systems have been actively studied. A telemedicine system capable of ECG measurement transmits ECG data in real time for real-time protection and analysis of heart-disease patients, and for this purpose, Bluetooth, communication mobile telephony network, and wireless local area network technologies are introduced [21–23]. Occlusion in one of the coronary arteries of the heart leads to cardiac ailment and myocardial infarction (MI). The localization of MI based on the investigation of the morphology of the multi-lead ECG is the initial task for the diagnosis of this ailment [24]. These telemedicine systems for constant observation of coronary heart disease patients provide the possibility for specialists to interpret ECGs on mobile devices, bridging the gap between patients and specialists [25]. The automated external defibrillator, which is used urgently in the event of a heart attack, analyzes the patient’s heart rhythm by measuring ECG signals to perform accurate defibrillation before the defibrillation. Additionally, after cardiac resynchronization therapy, ECG signals are measured to analyze the patient’s ECG pattern [26,27].

Chapter 2.2. R peak detection

Minor comments:

There are equations for the median filter (1) and adaptive filter size determination rules (2). It should be declared an equation for the difference signal as well. Of course, it is clear to see it from Figure 2, but the equation would give a complete definition of the entire process.

Ans) The following content and Equation (3) was added.

(Line 118) Additionally, the variance is obtained for the basic filter size. The variance section that determines the variable filter size is empirically divided into three uniform sections. The difference signal for the nth sliding window is given as follows:

.

(3)

Major comments:

From my perspective, the most important comment to this part is

the question, what is the background for the build of rules for defining the size of the adaptive filter? There are no references for previous works, and there is no description of how authors created the rules. Thus, it looks like magic. Why the authors use constants one third and two thirds? Moreover, the description of the rules it seems to me very brief. It would be suitable to consider depicted the process in the figure.

Ans) The following content and Figure (2) was added.

(Line 108) In the proposed method, a median filter with a variable filter size is used to improve the performance of a general median filter [36,37]. Figure 2 shows the relationship between the variance at the sample point and the median-filter size. A median filter with a large filter size is applied to a QRS section having a high variance, while a median filter with a small filter size is applied to a non-QRS section having a low variance.

Figure 2. Relationship of variance and filter size.

Chapter 2.3. Detection of Important Arrhythmias

Minor comments:

Line 106: However, Bigeminy ... It should be small "b".

Ans) It was corrected.

Line 107: in addition, ... It should be capital "I".

Ans) It was corrected.

Figure 3a: The description of TS is not right. TS is not the time, because there is multiplying of the Particle time and Sampling rate (there is no unit). The number gives the count of samples in the Sliding window. 

Ans) It was corrected to TS = Tp*N_Tp in Figure 4(a).

Figure 4: There should be mentioned initialization of N (N=1) at the start of the algorithm.

Ans) N=1 was added in Figure 5.

Major comments:

There is the same comment as in the previous chapter 2.2. There is no defined reason why the Particle time is 4s, the step size of 8s, the overlap is 4s and why there are 3 Particle times in the Sliding window. It should be mentioned why we cannot use 2s Particle time, 4s Sliding window or something else. If it is an empirical result from experiments, there should be some text about the optimization of these parameters. In this form, it looks like information from manual not from a scientific paper.

Ans) The following content was added.

(Line 153) The above sliding-window structure was derived manually to detect the arrhythmia presented in Table 1 in 1-min increments (every minute). Another sliding-window structure for detecting arrhythmia in units of 1 min may consist of Tp = 6 s, N_Tp = 4, and NSB = 3. Naturally, if the detection of arrhythmia in 1-min increments is ignored, various sliding-widget structures can be generated.

Figure 4: If I understand right, the arrhythmia detection process defines the Sliding window, and within each Sliding window, the R peak detection algorithm defines specific filter windows with different sizes. In other words, two processes use different windows in different time of processing and always the filter windows are defined within the currently used Sliding window. Is my understanding correct?

Ans) That’s right.

Chapter 2.3.2 Bigeminy and trigeminy

Minor comments:

Lines 137 and 138: ... represent the average length of the short, middle and long period in the observation time.

Ans) It was corrected.

Major comments:

Lines 144 and 145: Lines 144 and 145: Why the DiffLS and DiffMS are determined to 10? What was the process behind the determination of these values?

Ans) The following content was added.

(Line 183) In fact, these thresholds are very small values ​​for patients with bigeminy or trigeminy because the number of periods is very similar. Regarding Equation (5), if the average difference between the long and short periods within the observation time exists between the minimum and maximum differences, and the difference in the number of two periods is less than the threshold, the observed ECG signal is determined to be bigeminy.

Lines 149 and 150: The sentence does not describe the rule for a trigeminy recognition defined in equation 4.

Ans) The following content was added.

(Line 187) For an ECG signal that turns out to be bigeminy, additionally, if the average of the middle periods (actually, the average of the remaining periods excluding the short and long periods) exists between the averages of the short and long periods and the number of middle periods is similar to the number of short periods (or the number of long periods), the observed ECG signal is recognized as trigeminy.

Based on Figure 5: I will omit the question of what R peak is, whether R peak can be negative. I am not a physician, but I know that this question is no easy. In the depicted PVC beat cases, the negative peak can be probably rS or QS. However, it is just the question of the terminology. Let say that there is something like reverse R peak or mirrored peak. My question is how the algorithm can detect the negative peak. It means that there is absolute value in the calculation the difference signal. It is the next reason why to define the difference signal calculation by the equation. 

Ans) The following content was added.

(Line 130) Due to the absolute value characteristic of the difference signal in Equation (3), the R peak detected from the difference signal should be distinguished into a normal and abnormal R peak as follows.

(4)

If the detected R peak is greater than the average of the ECG signal of the current nth sliding window, it means a normal (positive) R, otherwise it is recognized as an abnormal R (negative R or PVC).

Chapter 2.3.3. PVC, VT, and SVT

Minor comments:

Figure 6: The legends are not exactly correct. In all figures, the signal is ECG not PVC, VT or SVT.

Ans) Figure 6 and 7 were revised based on your advice.

Chapter 2.3.4. AF

Minor comments:

Line 168: ... less than 120 ms wide ...

Ans) It was revised to “ <120 ms ”.

Lines 185 and 186: "... component of the fibrillatory wave can be detected using spectrogram or Short-time Fourier Transform." However, the spectrogram is just the visualization of the transformation from the time domain into the time-frequency domain, such as the Short-time Fourier Transform. Authors should reformulate this sentence.

Ans) It was revised as the following.

(Line 227) The high-frequency component of the fibrillatory wave can be detected using Short-time Fourier Transform.

Major comments:

Based on Figure 7: The spectrogram in the left part of the figure has defined frequency axis (y-axis) from 0 up to 5 Hz. It is a visualization of the high-frequency component of fibrillatory waves. However, the fibrillatory waves have the atrial rate from 300 to 600 waves per minute. In simple calculation (atrial rate/60), we obtain frequency 5 to 10 Hz. So, the picture is misleading. However, this is not my main point. Because, authors defined the normalizing OSC values correctly in the settings of the calculation (OSCmax = 15, OSCmin = 0), but there is again no explanation of how the values were acquired. It is same for other normalizing values in the AF features calculation process.

Ans) The following content was added and the spectrogram of Figure 8 was revised.

(Line 243) And the fibrillatory waves have the atrial rate from 300 to 600 waves per minute with varying morphology and high frequency [47]. The limit of the high waves of the fibrillation wave is occasionally set up to 650 [48]. Taking into account the waves per minute of the fibrillation wave, the min and max values ​​of the OSC were set.

Chapter 3. Results

The main problem with results represents not always a clear and reliable selection of tested signals.

Chapter 3.1.

Major comments:

The R peak detection was detected only on MITDB and only on 20 signals from the MITDB. The reason why the authors use only the part of one database is not declared. It is not necessary to show all results for all databases. However, it would be suitable to show at least the average results and not only for selected records but also for all record in the database. This selection of records leads to thinking that the authors selected only those records where the results were applicable for the paper. It is simply not an acceptable procedure. Authors declared in the chapter that they used Record 105 for comparison with other works because it is used in the larger number of works. This part is ok. However, it is not the reason why not show and comment on the results of R peak detection on all databases and all records.

Ans) All records of MITDB have been simulated as shown in Table 2 and the following content was added.

(Line 269) The detection performance for MITDB is shown in Table 2. An average QRS detection rate of 99.62%, a sensitivity of 99.82% and a positive prediction of 99.80% are obtained against all recordings of MITDB.

Chapter 3.2. Arrhythmia feature detection using various ECG data

Major comments:

There are just several pictures with some results, but there is no summary of results in this case. In fact, record 114 from the MITDB has the result of detection in the previous chapter. However, the rest of the records is without detection evaluation. They are simulated signals and there is not clear an obstacle to obtaining the detection evaluation. Moreover, there is not clear how many beats/signals were simulated and how many differences the signals contained (how was different variability of signals).

Ans) The following content was added.

(Line 308) Table 4 shows the performance evaluation of the arrhythmias tested in Figure 11. The evaluation of MITDB 114 was excluded because it is already indicated in Table 2. There was no false R detection for all arrhythmias (FP=0). While there were R peaks that were not detected in bigeminy and trigeminy, the R peaks of VT and SVT were almost completely detected. This is due to the non-detection of the weakly negative R peak at bigeminy and the lower two R peaks at trigeminy in one sliding window. It can be seen that the R peaks of bigeminy and trigeminy are not easy to detect completely due to R peaks of various heights.

Chapter 3.2. HRV analysis of various ECG signals

Minor comments:

The number of the chapter is wrong. It should be chapter 3.3.

Ans) It was corrected.

Figure 11: There is no a,b,c,d,e marking in the figure.

Ans) The marking was deleted in the caption.

Table 4: There is not clear why some values are bold.  Is it statistical significance? There is no mentioned test in the text.

Ans) It was corrected.

Major comments:

In fact, this chapter is a case study because there is used only one record for each type of signal (there is not distinguished that signals are simulated). Therefore, the results cannot be generalized. Table 4 and Figure 11 are just examples of how the HRV analysis parameters and the five features for AF detection can be distributed in specific cases of the signal. However, the generalization of the approach is zero because there is no statistical evaluation of the parameters and features/classification. It is the first step in the using of HRV for detection of other arrhythmias than only AF.  However, without evaluation, it is not so strong how it should be. Finally, there is not clear how authors intend the combination of HRV parameters and defined five features.

Ans) Figure 13 and 14, Table 5 were revised. And the following content was added.

(Line 363) Excluding records with problems with data length and data loading, the partial records of AFDB were used to extract the AF features. In these signals, the normal beat period and the AF period were alternately repeated. In the AF signal used in the experiment, a length of 24 min after 46 s was used, and Figure 13 shows the beginning 12 s of these AF signals. Figure 14 presents the normalized feature values ​​of the AF signals in polar coordinates. Figures 14(a) and 14(b) show the visualization of VT and SVT using AF detection technology. It can be seen that arrhythmias other than AF do not show a large area in AF visualization. Also, the normal signal will have a very small area in its visualization. As shown, more severe AF corresponded to larger AF feature values ​​in the polar coordinates. The severity of the AF is visually displayed. Records 04126 and 06995 exhibited better HRs. Record 06426 and 07161 show the concentrations of the RR intervals in the poincaré plot than the other signals. Record 06426 and 08378 were considered to be the most severe AF visually. For Records 07162, the high-frequency component in the spectrogram was small. Record 08405 exhibited better AA than the other signals.

(Line 376) Table 5 presents the five normalized feature values ​​for the AF signals. Among the AF signals used in the experiment, Record 04043 for HR, Record 04746, 06995, 07162,07910, and 08215 for AA, Record 06426 and 07162 for CPHB and SD, and Record 06426 and 08378 for OSC exhibited the highest (worst) feature values. An AF analysis revealed that the AF severity was in the following order: Records 06426, 08378, 06995, 05121, 04126, 04043, 07162, 05261, 08405, 05091, 07910, 07879, 04746, and 08215. The test AF signals used in the experiment show high average values ​​from 0.61 to 0.91 in normalized AF features. In addition, each normalized AF feature value for test AF signals shows a range of 0.63 to 0.94. It can be inferred that the proposed features can be used for AF detection through these experiments.

Chapter 3.3. AF detection

Minor comments:

The number of the chapter is wrong. It should be chapter 3.4.

Ans) It was corrected.

Line 308: There is no mentioned record 05121 in the selection of records from AFDB.

Ans) The paragraph was rewritten.

Table 5: There is not clear why different colors highlight several values.

Ans) Table 5 was corrected.

Major comments:

There is no clear why authors selected only six records of AFDB from 25 total.

Ans) Further simulation was performed and the part was newly rewritten (Line 363~384).

The problem is the same as in the previous chapter. There are observable differences between the features' representations. The visualization of arrhythmias seems to be promising. However, there are given results of specific signals always only from one subject.

Without statistical evaluation, there is impossible to evaluate the influence of the variability among records from different patients.

Ans) Further simulation was performed and the part was newly rewritten (Line 363~384).

Thank you for your valuable review !!

Reviewer 2 Report

The manuscript entitled “Adaptive Median Filter Based on Sampling Rate for 2 R peak detection and Major Arrhythmia Analysis” proposed an R point detection method relying on an adaptive median filter based on sampling rate. This method allows to analyze major arrhythmias from ECG signal features. The paper is interesting however some concerns have to be addresses:

MAJOR

1) The Figure often contains different panels that are not described in the main text (e.g. figure 2, 3, 5, 8, 9,11) or in the caption (e.g. figure 6). Figure 1 needs a more detailed explanation in the caption. From the figure it is not clear if also for real data acquired on participants a Matlab simulator is used. Moreover, no information regarding the participants of the study are reported in the manuscript (e.g. number of participants, age, gender, cardiac or circulatory diseases). The caption associated with figure 7 has to be more explicative in order to better understand the manuscript.

2) In the introduction section, some references regarding the employment of adaptive filters for the ECG data analysis should be added, in order to better compare the results with the state-of-the-art. For instance, you can refer to:

  • Weiting, Y., & Runjing, Z. (2007, August). An improved self-adaptive filter based on LMS algorithm for filtering 50Hz interference in ECG signals. In 2007 8th International Conference on Electronic Measurement and Instruments(pp. 3-874).
  • Hossain, M. B., Bashar, S. K., Walkey, A. J., McManus, D. D., & Chon, K. H. (2019). An Accurate QRS Complex and P Wave Detection in ECG Signals Using Complete Ensemble Empirical Mode Decomposition with Adaptive Noise Approach. IEEE Access7, 128869-128880.
  • Rahul, J., & Sora, M. (2020). A novel adaptive window based technique for T wave detection and delineation in the ECG. Bio-Algorithms and Med-Systems16(1).
  • Kurniawan, A., Yuniarno, E. M., Setijadi, E., Yusuf, M., & Purnama, I. K. E. (2020, July). QVAT: QRS Complex Detection based on Variance Analysis and Adaptive Threshold for Electrocardiogram Signal. In 2020 International Seminar on Intelligent Technology and Its Applications (ISITIA)(pp. 175-179).

MINORS:

1) A more detailed explanation regarding the choice of the parameters of the adaptive filter and the detection of the Atrial fibrillation has to be reported.

2) Figure 10 is not mentioned in the manuscript. The quality of figure 11 has to be improved. Figure 12 is missing.

3) The representation of AF signals on polar coordinates needs to be better explained. For instance, a comparison between a representation of AF with a healthy control signal should be reported in order to visualize the strength of this kind of representation.

4) In the Discussion section it would be worth to further stress the advantages of this approach. For instance, it could be used in those application where a real-time analysis is requested, such as continuous monitoring of the heartbeat or parameters that are related to cardiac rhythm (e.g., blood pressure), useful for the detection of cardiovascular diseases. For instance, you can refer to:

  • Lazazzera, R., Belhaj, Y., & Carrault, G. (2019). A new wearable device for blood pressure estimation using photoplethysmogram. Sensors, 19(11), 2557.
  • Xiao, N., Yu, W., & Han, X. (2020). Wearable heart rate monitoring intelligent sports bracelet based on Internet of things. Measurement, 164, 108102.
  • Bjerkne Wenneberg, S., Löwhagen Hendén, P. M., Oras, J., Naredi, S., Block, L., Ljungqvist, J., & Odenstedt Hergès, H. (2020). Heart rate variability monitoring for the detection of delayed cerebral ischemia after aneurysmal subarachnoid hemorrhage. Acta Anaesthesiologica Scandinavica.
  • Perpetuini, D., Chiarelli, A. M., Maddiona, L., Rinella, S., Bianco, F., Bucciarelli, V., ... & Fallica, G. (2019). Multi-site photoplethysmographic and electrocardiographic system for arterial stiffness and cardiovascular status assessment. Sensors, 19(24), 5570.

Author Response

# Reviewer 2

Comments and Suggestions for Authors

The manuscript entitled “Adaptive Median Filter Based on Sampling Rate for 2 R peak detection and Major Arrhythmia Analysis” proposed an R point detection method relying on an adaptive median filter based on sampling rate. This method allows to analyze major arrhythmias from ECG signal features. The paper is interesting however some concerns have to be addresses:

Ans) The whole manuscript was edited by native speaker.

MAJOR

1) The Figure often contains different panels that are not described in the main text (e.g. figure 2, 3, 5, 8, 9,11) or in the caption (e.g. figure 6). Figure 1 needs a more detailed explanation in the caption. From the figure it is not clear if also for real data acquired on participants a Matlab simulator is used. Moreover, no information regarding the participants of the study are reported in the manuscript (e.g. number of participants, age, gender, cardiac or circulatory diseases). The caption associated with figure 7 has to be more explicative in order to better understand the manuscript.

Ans) The figures and descriptions indicated have been modified. Additional explanation in Figure 1 has been added as follows.

(Line 92) MITDB is used to evaluate the R point detection method proposed in this paper.

(Line 96) AFDB is used to test the AF detection technique proposed in this paper and visualize the severity of the test AF signals.

(Line 99) The bigeminy signal used in this paper was measured by the developed ECG patch.

The actual measured signal is bigeminy, and this is explained as follows.

(line 292) Bigeminy is the signal measured by VP-100.

The vertical frequency range of the spectrogram in Figure 8 has been modified, and the description for Figure 8 has been added as follows.

(Line 227) The high-frequency component of the fibrillatory wave can be detected using Short-time Fourier Transform.

(Line 243) And the fibrillatory waves have the atrial rate from 300 to 600 waves per minute with varying morphology and high frequency [49]. The limit of the high waves of the fibrillation wave is occasionally set up to 650 [50]. Taking into account the waves per minute of the fibrillation wave, the min and max values of the OSC were set.

2) In the introduction section, some references regarding the employment of adaptive filters for the ECG data analysis should be added, in order to better compare the results with the state-of-the-art. For instance, you can refer to:

  • Weiting, Y., & Runjing, Z. (2007, August). An improved self-adaptive filter based on LMS algorithm for filtering 50Hz interference in ECG signals. In 2007 8th International Conference on Electronic Measurement and Instruments(pp. 3-874).
  • Hossain, M. B., Bashar, S. K., Walkey, A. J., McManus, D. D., & Chon, K. H. (2019). An Accurate QRS Complex and P Wave Detection in ECG Signals Using Complete Ensemble Empirical Mode Decomposition with Adaptive Noise Approach. IEEE Access7, 128869-128880.
  • Rahul, J., & Sora, M. (2020). A novel adaptive window based technique for T wave detection and delineation in the ECG. Bio-Algorithms and Med-Systems16(1).
  • Kurniawan, A., Yuniarno, E. M., Setijadi, E., Yusuf, M., & Purnama, I. K. E. (2020, July). QVAT: QRS Complex Detection based on Variance Analysis and Adaptive Threshold for Electrocardiogram Signal. In 2020 International Seminar on Intelligent Technology and Its Applications (ISITIA)(pp. 175-179).

Ans) The following content was added with the second and third reference papers.

(Line 47) And recently, QRS detection [21] and T-wave detection [22] using adaptive filters have been proposed.

MINORS:

1) A more detailed explanation regarding the choice of the parameters of the adaptive filter and the detection of the Atrial fibrillation has to be reported.

Ans) The following content and Figure (2) was added.

(Line 108) In the proposed method, a median filter with a variable filter size is used to improve the performance of a general median filter [36,37]. Figure 2 shows the relationship between the variance at the sample point and the median-filter size. A median filter with a large filter size is applied to a QRS section having a high variance, while a median filter with a small filter size is applied to a non-QRS section having a low variance.

Figure 2. Relationship of variance and filter size.

Figure 13 and 14, Table 5 were revised. And the following content was added.

(line 363) Excluding records with problems with data length and data loading, the partial records of AFDB were used to extract the AF features. In these signals, the normal beat period and the AF period were alternately repeated. In the AF signal used in the experiment, a length of 24 min after 46 s was used, and Figure 13 shows the beginning 12 s of these AF signals. Figure 14 presents the normalized feature values ​​of the AF signals in polar coordinates. Figures 14(a) and 14(b) show the visualization of VT and SVT using AF detection technology. It can be seen that arrhythmias other than AF do not show a large area in AF visualization. Also, the normal signal will have a very small area in its visualization. As shown, more severe AF corresponded to larger AF feature values ​​in the polar coordinates. The severity of the AF is visually displayed. Records 04126 and 06995 exhibited better HRs. Record 06426 and 07161 show the concentrations of the RR intervals in the poincaré plot than the other signals. Record 06426 and 08378 were considered to be the most severe AF visually. For Records 07162, the high-frequency component in the spectrogram was small. Record 08405 exhibited better AA than the other signals.

(Line 376) Table 5 presents the five normalized feature values ​​for the AF signals. Among the AF signals used in the experiment, Record 04043 for HR, Record 04746, 06995, 07162,07910, and 08215 for AA, Record 06426 and 07162 for CPHB and SD, and Record 06426 and 08378 for OSC exhibited the highest (worst) feature values. An AF analysis revealed that the AF severity was in the following order: Records 06426, 08378, 06995, 05121, 04126, 04043, 07162, 05261, 08405, 05091, 07910, 07879, 04746, and 08215. The test AF signals used in the experiment show high average values ​​from 0.61 to 0.91 in normalized AF features. In addition, each normalized AF feature value for test AF signals shows a range of 0.63 to 0.94. It can be inferred that the proposed features can be used for AF detection through these experiments.

2) Figure 10 is not mentioned in the manuscript. The quality of figure 11 has to be improved. Figure 12 is missing.

Ans) Figure 11 was mentioned (Line 290). The quality of figure 12 was revised. The figure number was revised.

3) The representation of AF signals on polar coordinates needs to be better explained. For instance, a comparison between a representation of AF with a healthy control signal should be reported in order to visualize the strength of this kind of representation.

Ans) Further simulation was performed and the part was newly rewritten (Line 359~380).

4) In the Discussion section it would be worth to further stress the advantages of this approach. For instance, it could be used in those application where a real-time analysis is requested, such as continuous monitoring of the heartbeat or parameters that are related to cardiac rhythm (e.g., blood pressure), useful for the detection of cardiovascular diseases. For instance, you can refer to:

  • Lazazzera, R., Belhaj, Y., & Carrault, G. (2019). A new wearable device for blood pressure estimation using photoplethysmogram. Sensors, 19(11), 2557.
  • Xiao, N., Yu, W., & Han, X. (2020). Wearable heart rate monitoring intelligent sports bracelet based on Internet of things. Measurement, 164, 108102.
  • Bjerkne Wenneberg, S., Löwhagen Hendén, P. M., Oras, J., Naredi, S., Block, L., Ljungqvist, J., & Odenstedt Hergès, H. (2020). Heart rate variability monitoring for the detection of delayed cerebral ischemia after aneurysmal subarachnoid hemorrhage. Acta Anaesthesiologica Scandinavica.
  • Perpetuini, D., Chiarelli, A. M., Maddiona, L., Rinella, S., Bianco, F., Bucciarelli, V., ... & Fallica, G. (2019). Multi-site photoplethysmographic and electrocardiographic system for arterial stiffness and cardiovascular status assessment. Sensors, 19(24), 5570.

 Ans) The following content was added with the first and second reference papers.

(Line 402) The proposed R detection and HRV analysis technology can be directly utilized for real-time heart rate monitoring through a wearable ECG measuring device [58]. In addition, HRV parameters related to autonomic and non-autonomous nervous systems can be used as additional parameters for blood pressure estimation using photoplethysmography [59].

Thank you for your valuable review !!

Round 2

Reviewer 1 Report

Comments:

Chapter 1: Introduction
Lines 51-53: "A telemedicine system capable of ECG measurement transmits ECG data in real time for real-time 52 protection and analysis of heart-disease patients, and for this purpose, Bluetooth, communication mobile telephony network, and wireless local area network technologies are introduced [23–25]."
I am not a native speaker, and maybe, therefore, this sentence is quite challenging read for me. However, that is just a small comment.

Chapter 2.1. ECG Data Used
I appreciate adding o Figure 2. However, first, there is a discrepancy between named of filter size. In the text, it is a large filter size (correct) in the figure it is "Big filter size". Second, there is no clear how the filter size is related to the variance because a small window depicts a large filter size (QRS). I know that the window does not represent the filter size, but, from the figure, it seems to be so.

Chapter 2.3.4. AF
Unfortunately, the authors picked only the one case of the parameters setting that I mentioned in my previous review. Why BPMmax = 100, MSDmax =0.01, and CPHBmax = 1 is still without any reason. However, these parameters are not crucial for the intention of the paper.

Chapter 3.2. Arrhythmic Feature Detection Using Various ECG Data

Line 333: The labelling of tables starts to be misleading. Table 4 is on page 13. Next, there are tables 3 and 4 on pages 14 and 15. Finally, table 5 is on pages 18 and 19. It must be corrected.

Author Response

Chapter 1: Introduction
Lines 51-53: "A telemedicine system capable of ECG measurement transmits ECG data in real time for real-time 52 protection and analysis of heart-disease patients, and for this purpose, Bluetooth, communication mobile telephony network, and wireless local area network technologies are introduced [23–25]."
I am not a native speaker, and maybe, therefore, this sentence is quite challenging read for me. However, that is just a small comment.

Ans) We revised the sentence.

(Line 51) Using Bluetooth, communication mobile telephony network, and wireless local area network technologies, recent telemedicine systems measure and transmit ECG data in real-time to protect heart-disease patients [23–25].

Chapter 2.1. ECG Data Used
I appreciate adding o Figure 2. However, first, there is a discrepancy between named of filter size. In the text, it is a large filter size (correct) in the figure it is "Big filter size". Second, there is no clear how the filter size is related to the variance because a small window depicts a large filter size (QRS). I know that the window does not represent the filter size, but, from the figure, it seems to be so.

Ans) We revised Figure 2 and the following sentences.

(Line 109~114) Figure 2 shows the relationship between the variance at the sample point and the median-filter size for AFDB of 250 samples/s. A median filter with a large filter size is applied to QRS interval having a high variance, while a median filter with a small filter size is applied to flat region (or weak wave region) among non-QRS interval having a low variance. Also, a median filter with a basic size is applied to the P or T wave region among the non-qrs region corresponding to the moderate variance.

(Line 118) The adaptive size of the median filter for a sampling rate of 250 Hz is shown in Figure 2.

Figure 2. Relationship of variance and filter size: an example of 250 samples/s.

Chapter 2.3.4. AF
Unfortunately, the authors picked only the one case of the parameters setting that I mentioned in my previous review. Why BPMmax = 100, MSDmax =0.01, and CPHBmax = 1 is still without any reason. However, these parameters are not crucial for the intention of the paper.

Ans) We added the following explanation.

(Line 246~250) The max of the BPM is set based on tachycardia, which means more than 100 beats per minute. One standard deviation value of 0.1 was determined assuming that the difference between the current and next RR intervals for a stable heart rate was 0.1 or less. As a result, the max of the MSD was set at 0.01. Since the PHB is a probability between 0 and 1, the CPHB also has a value in the same range.

Chapter 3.2. Arrhythmic Feature Detection Using Various ECG Data

Line 333: The labeling of tables starts to be misleading. Table 4 is on page 13. Next, there are tables 3 and 4 on pages 14 and 15. Finally, table 5 is on pages 18 and 19. It must be corrected.

Ans) We revised the errors concerning numbering of Tables and Figures.

Reviewer 2 Report

The Paper is improved after the revision and in my opinion it is suitable for publication in the present form.

Author Response

The Paper is improved after the revision and in my opinion it is suitable for publication in the present form.

Thank you for your valuable review !!
